# Simplify and Robustify Negative Sampling for Implicit Collaborative Filtering

**Jingtao Ding**[1][*]   **Yuhan Quan**[1]   **Quanming Yao**[1,2]   **Yong Li**[1]   **Depeng Jin**[1]

[1]Department of Electronic Engineering, Tsinghua University,
[2]4Paradigm Inc.

## Abstract

Negative sampling approaches are prevalent in implicit collaborative filtering for obtaining negative labels from massive unlabeled data. As two major concerns in negative sampling, efficiency and effectiveness are still not fully achieved by recent works that use complicate structures and overlook risk of false negative instances. In this paper, we first provide a novel understanding of negative instances by empirically observing that only a few instances are potentially important for model learning, and false negatives tend to have stable predictions over many training iterations. Above findings motivate us to simplify the model by sampling from designed memory that only stores a few important candidates and, more importantly, tackle the untouched false negative problem by favouring high-variance samples stored in memory, which achieves efficient sampling of true negatives with high-quality. Empirical results on two synthetic datasets and three real-world datasets demonstrate both robustness and superiorities of our negative sampling method. The implementation is available at `https://github.com/dingjingtao/SRNS`.

## 1 Introduction

Collaborative filtering (CF), as the key technique of personalized recommender systems, focuses on learning user preference from the observed user-item interactions [28, 34]. Today's recommender systems also witness the prevalence of implicit user feedback, such as purchases in E-commerce sites and watches in online video platforms, which is much easier to collect compared to the explicit feedback (such as ratings) on item utility. In above examples, each observed interaction normally indicates a user's interest on an item, *i.e.*, a positive label, while the rest unobserved interactions are unlabeled. As for learning an implicit CF model from this positive-only data, a widely adopted approach is to select a few instances from the unlabeled part and treat them as negative labels, also known as negative sampling [10, 34]. Then, the CF model is optimized to give positive instances higher scores than those given to negative ones [34].

Similar to other related applications in representation learning of text [27] or graph data [30], negative sampling in implicit CF also has two major concerns, *i.e.*, efficiency and effectiveness [10, 46]. First, the efficient sampling process is required, as the number of unobserved user-item interactions can be extremely huge. Second, the sampled instances need to be high-quality, so as to learn useful information about user's negative preference. However, since implicit CF is an application-driven problem where user behaviors play an important role, it may be unrealistic to assume that unobserved interactions are all negative, which introduces false negative instances into training process [21, 26, 49]. For example, an item may be ignored because of its displayed position and form, not necessarily the user's dislike. Therefore, false negative instances naturally exist in implicit CF.

---

[*]The first three authors have equal contributions.

Table 1: Comparison of the proposed SRNS with closely related works, where $\text{rk}(j|u)$ is the $(u, j)$'s rank sorted by score, $\text{pop}_j$ is the $j$'s item popularity, $B$ is the mini-batch size, $T$ is the time complexity of computing an instance score, $E$ is the epoch of lazy-update, and $\mathcal{F}$ denotes false negative.

| | $p_{\text{ns}}(j|u)$ | Optimization | Time Complexity | Robustness |
|---|---|---|---|---|
| Uniform [34] | $\text{Uniform}(\{j \notin \mathcal{R}_u\})$ | SGD (from scratch) | $O(BT)$ | $\times$ |
| NNCF [10] | $\propto (\text{pop}_j)^{0.75}$ | SGD (from scratch) | $O(B^2T)$ | $\times$ |
| AOBPR [33] | $\propto \exp(-\text{rk}(j|u)/\lambda)$ | SGD (from scratch) | $O(BT)$ | $\times$ |
| IRGAN [38] | learned $\bar{p}_{\text{ns}}(j|u)$ (GAN) | REINFORCE (pretrain) | $O(B|\mathcal{I}|T)$ | $\times$ |
| AdvIR [29] | learned $\bar{p}_{\text{ns}}(j|u)$ (GAN) | REINFORCE (pretrain) | $O(BS_1T)$ | $\times$ |
| SRNS (proposed) | variance-based (see (4)) | SGD (from scratch) | $O(\frac{B}{E}(S_1 + S_2)T)$ | $\checkmark$ |

Previous works of negative sampling in implicit CF mainly focus on replacing the uniform sampling distribution with another proposed distribution, so as to improve the quality of negative samples. Similar to the word-frequency based distribution [27] and node-degree based distribution [30] used in other domains, an item-popularity based distribution that favours popular items is usually adopted [10, 42]. In terms of sample quality, the strategy emphasizing hard negative samples has been proven to be more effective [29], as it can bring more information for model training. Here the hard samples refer to those with a high probability of being positive according to the model, which are hard for learning. Specifically, this is achieved by either assigning higher probability to instances with large prediction score [33, 46] or leveraging techniques of adversarial learning [12, 29, 38]. Nevertheless, the above hard negative sampling approaches cannot simultaneously meet the requirements on efficiency and effectiveness. On the one hand, several state-of-the-art solutions [12, 29] use complicate structures like generative adversarial network (GAN) [18] for generating negative instances, which has posed a severe challenge on model efficiency. On the other hand, all these methods overlook the risk of introducing false negative instances and instead only focus on hard ones, making the sampling process less robust for training an effective CF model with false negatives.

Different from above works, this paper formulates the negative sampling problem as efficient learning from unlabeled data with the presence of noisy labels, *i.e.*, false negative instances. We propose to simplify and robustify the negative sampling for implicit CF, which has three main challenges:

- **How to capture the distribution of true negative instances with an unbiased but simple model?** In the implicit CF problem, true negative instances are hidden inside the massive unlabeled data, along with false negative instances. Thus negative sampling for implicit CF expects an unbiased estimator that correctly identifies true negative instances during training process. On the other hand, previous works have shown that negative instances in other domains follow a skewed distribution and can be modeled by a simple model [27, 47]. However, it remains unknown in the implicit CF problem if this prior knowledge can also help building an unbiased but simple model for negative sampling.
- **How can we reliably measure the quality of negative samples?** Given the risk of introducing false negative instances, the quality of negative samples needs to be measured in a more reliable way. However, it is non-trivial to design a discriminative criterion that can help to accurately identify true negative instances with high quality.
- **How can we efficiently sample true negative instances of high-quality?** Although learning effective information from unlabeled and noisy data is related to general machine learning approaches including positive-unlabeled leaning [23] and instance re-weighting [32], these methods are not suitable for implicit CF problem, where the huge number of unobserved user-item interactions requires an efficient modeling. Instead, our proposed method needs to maintain both efficiency, by sampling, and effectiveness, by considering samples' informativeness and reliability simultaneously. This has not been tackled before in both implicit CF and other similar problems.

Solving above three challenges calls for a deep and fundamental understanding of different negative instances in implicit CF problem. In this paper, we empirically find that negative instances with large prediction scores are important for the model learning but generally rare, *i.e.*, following a skewed distribution. A more novel finding is that false negative instances always have large scores over many iterations of training, *i.e.*, a lower variance, which provides a new angle on tackling false negative problem remained in existing approaches. Motivated by above two findings, we

propose a novel simplified and robust negative sampling approach, named SRNS, that 1) captures the dynamic distribution of negative instances with a memory-based model, by simply maintaining the promising candidates with large scores, and 2) leverages a high-variance based criterion to reliably measure the quality of negative samples, reducing the risk of false negative instances effectively. Above two designs are further combined into a two-step sampling scheme that constantly alternates between score-based memory update and variance-based sampling, so as to efficiently sample true negative instances with high-quality. Experiment results on two synthetic datasets demonstrate the robustness of our SRNS under various levels of noisy circumstances. Further experiments on three real-world datasets also empirically validates its superiorities over state-of-the-art baselines, in terms of effectiveness and efficiency.

## 2 Background

Training an implicit CF model generally involves three main steps, *i.e.*, choosing scoring function $r$, objective function $L$ and negative sampling distribution $p_{\mathrm{ns}}$. The scoring function $r(\mathbf{p}_u, \mathbf{q}_i, \boldsymbol{\beta})$ calculates the relevance between a user $u \in \mathcal{U}$ and an item $i \in \mathcal{I}$ based on $u$'s embedding $\mathbf{p}_u \in \mathbb{R}^F$ and $i$'s embedding $\mathbf{q}_i \in \mathbb{R}^F$, with a learnable parameter $\boldsymbol{\beta}$. It can be chosen among various candidates including matrix factorization (MF) [24], multi-layer perceptron (MLP) [20], graph neural network (GNN) [3, 40], etc. For example, the generalized matrix factorization (GMF) [20] is: $r(\mathbf{p}_u, \mathbf{q}_i, \boldsymbol{\beta}) = \boldsymbol{\beta}^\top (\mathbf{p}_u \odot \mathbf{q}_i)$, where the learnable parameter of $r$ is a vector $\boldsymbol{\beta}$ and $\odot$ denotes element-wise product. A large value of $r(\mathbf{p}_u, \mathbf{q}_i, \boldsymbol{\beta})$ indicates $u$'s strong preference on $i$, denoted by $r_{ui}$ for simplicity. Each observed instance between $u$ and the interacted item $i \in \mathcal{R}_u$, *i.e.*, $(u, i)$, can be seen as a positive label. As for the rest unobserved interactions, *i.e.*, $\{(u, j)|j \notin \mathcal{R}_u\}$, the probability of $(u, j)$ being negative is

$$P_{\mathrm{neg}}(j|u, i) = \mathrm{sigmoid}(r_{ui} - r_{uj}), \tag{1}$$

which approaches to 1 when $r_{ui} \gg r_{uj}$. In other words, when learning user preference in implicit CF, we care more about the pairwise ranking relation between an observed interaction $(u, i) \in \mathcal{R}$ and another unobserved interaction $(u, j)$, instead of absolute values of $r_{ui}$ and $r_{uj}$. The learning objective can be formulated as minimizing following loss function [34]:

$$L(\{\mathbf{p}_u\}, \{\mathbf{q}_i\}, \boldsymbol{\beta}) = \sum\nolimits_{(u,i) \in \mathcal{R}} \left[ \mathbb{E}_{j \sim p_{\mathrm{ns}}(j|u)} \left[ -\log P_{\mathrm{neg}}(j|u, i) \right] \right], \tag{2}$$

where the negative instance $(u, j)$ is sampled according to a specific distribution $p_{\mathrm{ns}}(j|u)$. Learning above objective is equivalent to maximizing the likelihood of observing such pairwise ranking relations $r_{ui} > r_{uj}$, which can be replaced by other objectives used in implicit CF problems, such as marginal hinge loss [44] and binary cross-entropy loss [20].

The most widely used $p_{\mathrm{ns}}(j|u)$ is the uniform distribution [34], suffering from low quality of samples. To solve this, previous works [12, 29, 33] propose to sample much harder instances, containing more information. Among them, state-of-the-arts [12, 29] simultaneously learn a parameterized $\bar{p}_{\mathrm{ns}}(j|u)$ to maximize above loss function in (2), based on GAN. Therefore, the sampled negative instance $(u, j)$ corresponds to a low $P_{\mathrm{neg}}(j|u, i)$ and a high $r_{uj}$, which is generally hard for CF model to learn. In other words, $(u, j)$ has a high probability of being positive, denoted as $P_{\mathrm{pos}}(j|u, i) = 1 - P_{\mathrm{neg}}(j|u, i)$. Different choices of $p_{\mathrm{ns}}(j|u)$ in previous works are listed in Table 1. Since none of them have enough robustness to handle false negative instances, and GAN-based model structure is much more complicate, our goal is to propose a more robust and simplified negative sampling method.

## 3 SRNS: the Proposed Method

To improve robustness and efficiency for negative sampling in implicit CF, we seek for a deep understanding of different negative instances, including false negative instances and negative instances obtained by uniform sampling or hard negative sampling. We then describe the proposed method based on these understandings.

### 3.1 Understanding False Negative Instances

In previous works [29, 33], the positive-label probability $P_{\mathrm{pos}}$ (or the prediction score) is widely used as the sampling criterion, as it is proportional to the sample difficulty. Therefore, in Figure 1 (details

on setup are in Appendix C.2), we have a closer look at the negative instances' distribution *w.r.t.* $P_{pos}$ and further analyze the possibility of using $P_{pos}$ to discriminate true negative instances and false negative instances. Besides, we are also curious about the model's prediction uncertainty regarding to different negative instances, and investigate the variance of $P_{pos}$ in Figure 1(d).

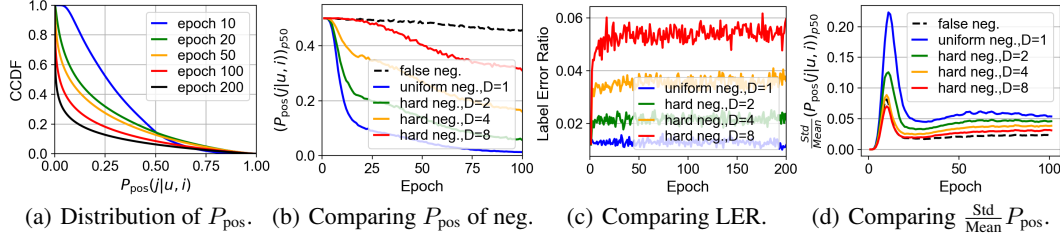

(a) Distribution of $P_{pos}$.    (b) Comparing $P_{pos}$ of neg.    (c) Comparing LER.    (d) Comparing $\frac{\text{Std}}{\text{Mean}}P_{pos}$.

Figure 1: Analysis of negative instances on ML-100k. $D$: difficulty level; Label Error Ratio (LER): $=$ (# of false negative samples)$/$(# of all selected negative samples); CCDF: complementary cumulative distribution function, $p50$: median value among a set of negative instances, $^{\text{Std}}/_{\text{Mean}}$: normalized variance).

Based on above analysis of negative instances in implicit CF, we have following two findings:

1) The score distribution of negative instances is highly skew. Regardless of the training, only a few negative instances have large scores (Figure 1(a)).
2) Both false negative instances and hard negative instances have large scores (Figure 1(b)), making it hard to discriminate them (harder negative samples are more likely be false negative, Figure 1(c)). However, the false negative instances have lower prediction variance comparatively (Figure 1(d)).

The first finding demonstrates the potential of just capturing a part of the full distribution corresponding to those large-scored negative instances, which are more likely to be high-quality. Similar observations have also been discussed in graph representation learning, suggesting a skewed negative sampling distribution that focuses on hard ones [43, 47].

As for the second finding, it provides us a reliable way of measuring sample quality based on prediction variance, sharing the same intuition with [8] that improves stochastic optimization by emphasizing high variance samples. Specifically, we prefer those negative samples with both large scores and high variances, avoiding false negative instances that always have large scores over many iterations of training. In terms of robustifying negative sampling process, none of above works in implicit CF and other domains have tackled this problem, except for a simple workaround that only selects hard negative samples but avoids the hardest ones [44, 47].

## 3.2 SRNS Method Design

As above, on the one hand, we are motivated to use a small amount of memory for each user, storing hard negative instances that have large potential of being high-quality. This largely simplifies the model structure, by focusing on a partial set of instances, which thus improves efficiency. On the other hand, we propose a variance-based sampling strategy to effectively obtain samples that are both reliable and informative. Our simplified and robust negative sampling (SRNS) approach addresses the remaining challenges on model efficiency and robustness. Algorithm 1 shows the implicit CF learning framework, *i.e.*, minimizing loss function in (2), based on SRNS.

The learning process of the SRNS is carried out in mini-batch mode, and alternates between two main steps. First, according to the high-variance based criterion, a negative instance for each training instance $(u, i)$ is sampled from $u$'s memory $\mathcal{M}_u$ (line 6), which already stores $S_1$ candidates with high potential. To improve efficiency, all positive instances of a same user $u$ is designed to share one memory $\mathcal{M}_u$. Second, as the model is constantly changing during the training process, $\mathcal{M}_u$ requires a dynamic update so as to keep track of the promising candidates for negative sampling. Specifically, this is completed by first extending it into $\mathcal{M}_u \cup \bar{\mathcal{M}}_u$ with additional $S_2$ instances that was uniformly sampled (line 7), and then choosing $S_1$ hard candidates to obtain a new $\mathcal{M}_u$ (line 8). A similar two-step scheme is adopted by a related work that focuses on negative sampling for knowledge graph embeddings [47]. However, unlike SRNS leveraging the instance's variance in the sampling step, it uniformly chooses an instance from memory, which cannot enhance model's robustness effectively.

**Algorithm 1:** The proposed Simplified and Robust Negative Sampling (SRNS) method.

---
**Input** : Training set $\mathcal{R} = \{(u, i)\}$, embedding dimension $F$, scoring function $r$ with learnable parameter $\boldsymbol{\beta}$, and memory $\{\mathcal{M}_u | u \in \mathcal{U}\}$, each with size $S_1$;
**Output** : Final user embeddings $\{\mathbf{p}_u | u \in \mathcal{U}\}$ and item embeddings $\{\mathbf{q}_i | i \in \mathcal{I}\}$, and $r$;

1  Initialize $\{\mathbf{p}_u | u \in \mathcal{U}\}$ and $\{\mathbf{q}_i | i \in \mathcal{I}\}$, $\boldsymbol{\beta}$ and $\{\mathcal{M}_u | u \in \mathcal{U}\}$;
2  **for** $t = 1, 2, ..., T$ **do**
3      Sample a mini-batch $\mathcal{R}_{batch} \in \mathcal{R}$ of size $B$;
4      **for** *each* $(u, i) \in \mathcal{R}_{batch}$ **do**
5          Get the candidate items from $u$-related memory $\mathcal{M}_u$;
6          Sample the item $j$ from $\mathcal{M}_u$, based on the variance-based sampling strategy (4);
7          Uniformly sample $S_2$ items from $\{k | k \notin \mathcal{R}_u\}$ ($\bar{\mathcal{M}}_u$), and merge with original $\mathcal{M}_u$;
8          Update $\mathcal{M}_u$ based on the score-based updating strategy (3);
9          Update embeddings $\{\mathbf{p}_u, \mathbf{q}_i, \mathbf{q}_j\}$ and parameters $\boldsymbol{\beta}$ based on gradient *w.r.t.* $L$ (2).
10     **end**
11 **end**

---

### 3.2.1 Score-based memory update

In this part, we propose a memory-based model to simply capture the dynamic distributions of true negative instances. Specifically, a memory $\mathcal{M}_u = \{(u, k_1), ..., (u, k_{S_1})\}$ of size $S_1$ is assigned to each user $u$, storing the negative instances that are available to $u$ in sampling. To ensure only those informative instances are maintained, we design a score-based strategy to dynamically update the $\mathcal{M}_u$, which tends to involve more hard negative instances. For an extended memory that merges the old $\mathcal{M}_u$ and a set of uniformly sampled instances $\bar{\mathcal{M}}_u$, *i.e.*, $\mathcal{M}_u \cup \bar{\mathcal{M}}_u$, the new $\mathcal{M}_u$ is updated by sampling $S_1$ instances according to the following probability distribution:

$$\bar{\Psi}(k|u, \mathcal{M}_u \cup \bar{\mathcal{M}}_u) = \exp(r_{uk}/\tau) / \sum\nolimits_{k' \in \mathcal{M}_u \cup \bar{\mathcal{M}}_u} \exp(r_{uk'}/\tau), \tag{3}$$

where a lower temperature $\tau \in (0, +\infty)$ would make $\bar{\Psi}$ focus more on large-scored instances.

### 3.2.2 Variance-based sampling

As we have demonstrated in finding 2, oversampling hard negative instances may increase the risk of introducing false negatives, making above score-based updating strategy less robust. Motivated by the observed low-variance characteristic of false negatives, we propose a robust sampling strategy that can effectively avoid this noise by favouring those high-variance candidates. Given a positive instance $(u, i)$ and $u$'s memory $\mathcal{M}_u$, for each candidate $(u, k) \in \mathcal{M}_u$, we maintain $P_{\text{pos}}(k|u, i)$ values at $t$th training epoch as $[P_{\text{pos}}(k|u, i)]_t$. The proposed variance-based sampling strategy chooses the negative instance $(u, j)$ from $\mathcal{M}_u$ by:

$$j = \arg\max\nolimits_{k \in \mathcal{M}_u} P_{\text{pos}}(k|u, i) + \alpha_t \cdot \text{std}[P_{\text{pos}}(k|u, i)]. \tag{4}$$

Note that we also consider the instance difficulty, i.e., $P_{\text{pos}}(k|u, i)$, to ensure the informativeness of sampled negative instances, with a hyper-parameter $\alpha_t$ controlling the importance of high-variance at the $t$-th training epoch. When $\alpha_t = 0$, our proposed sampling approach degenerates into a difficulty-only strategy that follows the similar idea as previous works [12, 29, 33]. Since all instances tend to have high variance at an early training stage, the variance term should not be weighted too much. Therefore, we expect a "warm-start" setting of $\alpha_t$ that reduces the influence of prediction variance at first and then gradually strengthens it (details of $\alpha_t$ will be discussed in Section 4.2).

For each candidate sample stored in memory $\mathcal{M}_u$, we directly use its corresponding prediction probability in the latest 5 epochs to compute the variance. Specifically, at $t$th training epoch,

$$\begin{aligned} \text{std}[P_{\text{pos}}(k|u, i)] &= \sqrt{\sum\nolimits_{s=t-5}^{t-1} \big[[P_{\text{pos}}(k|u, i)]_s - \text{Mean}[P_{\text{pos}}(k|u, i)]\big]^2 / 5}, \\ \text{Mean}[P_{\text{pos}}(k|u, i)] &= \sum\nolimits_{s=t-5}^{t-1} [P_{\text{pos}}(k|u, i)]_s / 5. \end{aligned} \tag{5}$$

In case that some samples may just enter $\mathcal{M}_u$, their prediction results will be tracked in our implementation, which requires extra memory cost and computation. However, this score tracking process only occurs once per iteration, and no backward computation is required, making its overhead with the same magnitude as that of the sampling operation (details are in Appendix B.6).

### 3.2.3 Bootstrapping

In Algorithm 1, false negative instances are identified by checking their prediction variance. However, this assumes that the CF model has some discriminative ability. There is an important observation that deep models can memorize easy training instances first and gradually adapt to hard instances [1, 19, 45]. Fortunately, we also observe such memorization effect for deep CF models (see Section 4.2), which means that the false negative instances among unlabeled data are generally more difficult and may not be memorized at an early stage. In other words, SRNS can be self-boosted by first learning to discriminate those easy negative instances and then tackling the rest real hard ones with the help of variance-based criterion.

### 3.3 Complexity Analysis

Here, we analyze the time complexity of SRNS (Algorithm 1) and compare it with related negative sampling approaches in Table 1. Compared with a uniform sampling approach [34], the main additional cost comes from score-based memory update and variance-based sampling. The former requires to compute scores of $S_1 + S_2$ candidates for each positive instance and sample $S_1$ of them according to a normalized distribution $\bar{\Psi}$ that is based on computed scores. Thus the time complexity is $O((S_1 + S_2)T)$, where $T$ denotes the operation count of score computation. As for the latter, computing $\text{std}[P_{\text{pos}}(k|u, i)]$ of each candidate and choosing the final negative instance in (4) take $O(S_1)$. Thus, the cost is $O((S_1 + S_2)T)$ for each positive instance, which can be reduced to $O((S_1 + S_2)T/E)$ using lazy-update every $E$ epochs. Model parameters in CF consists of two parts, *i.e.*, embeddings and scoring function parameters, and the former is generally much larger. Specifically, SRNS's model complexity is about $(|\mathcal{U}| + |\mathcal{I}|)F$, which can double in those GAN-based state-of-the-arts [29, 38]. As in Table 1, SRNS is not only more simplified (in terms of time complexity), but also can be easily trained from scratch.

## 4 Experiments

We first conduct controlled experiments with synthetic noise, so as to investigate SRNS's robustness to false negative instances (Section 4.2). Then, we evaluate the SRNS's performance on the implicit CF task, based on real data experiments (Section 4.3).

### 4.1 Experimental Settings

*Dataset.* Table 2 summarizes datasets used for experiments, which are popularly used in the literature [17, 20, 29, 38]. We use ML-100k and Ecom-toy for synthetic data experiments and do a 4:1 random splitting for train/test data. To simulate the noise, we randomly select 50%/25% of groundtruth records in the test set of ML-100k/Ecom-toy. The selected records can be regarded as false negative instances during training, denoted as $\mathcal{F}$. As for real data experiments, we use the rest three datasets and adopt leave-one-out strategy, *i.e.*, holding out users' most recent records and second to the last records (sorted *w.r.t.* time-stamp) as the test set and validation set, respectively [20, 34].

Table 2: Statistics of datasets.

| Category | Dataset | User | Item | Train | Validation | Test | $\mathcal{F}$ (Noise) |
|---|---|---|---|---|---|---|---|
| Synthetic noisy dataset | Movielens-100k | 942 | 1,447 | 44,140 | - | 11,235 | 5,509 |
| | Ecommerce-toy | 1,000 | 2,000 | 60,482 | - | 14,612 | 3,246 |
| Real-world dataset | Movielens-1m | 6,028 | 3,533 | 563,186 | 6,028 | 6,028 | - |
| | Pinterest | 55,187 | 9,916 | 1,390,435 | 55,187 | 55,187 | - |
| | Ecommerce | 66,450 | 59,290 | 1,625,006 | 66,441 | 66,450 | - |

*Baselines.* We compare SRNS with three types of methods listed in Table 1. First, for methods using a fixed negative sampling distribution, we choose Uniform [34] and NNCF [10]. Second, for methods based on hard negative sampling, we choose AOBPR [33], IRGAN [38], RNS-AS [12] and AdvIR [29], where the last three are GAN-based state-of-the-arts. Finally, we also compare with a non-sampling approach ENMF [9] that regards all the unlabeled data as negative labels.

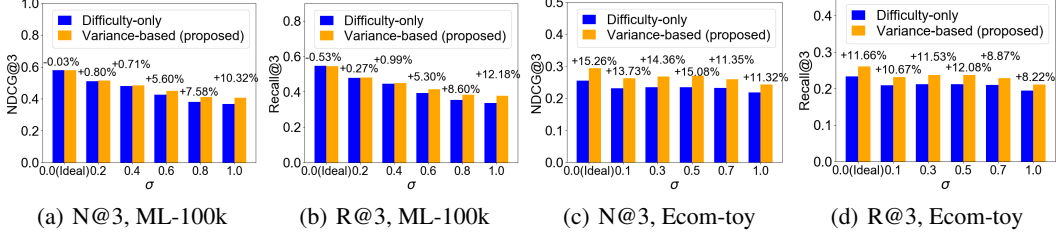

Figure 2: Average test Recall/NDCG of SRNS with different $\sigma$ on two synthetic datasets over the last 50 epochs. Two sampling strategies are compared, *i.e.*, difficulty-only vs. variance-based (proposed).

*Hyper-parameter and optimizer.* For better performance, we mainly use GMF [20] as the scoring function $r$, but also experiment on another popular choice, *i.e.*, a MLP with sigmoid activation (Section 4.3). Hyper-parameters of SRNS and baselines are carefully tuned according to validation performance (details are in Appendix B.4). For all experiments, Adam optimizer is used and the mini-batch size is 1024. Specifically, we run each synthetic data experiment 400 epochs and repeat five times. As for real data experiments, we conduct the standard procedure [10, 40], running 400 epochs and terminating training if validation performance does not improve for 100 epochs.

*Experimental setup.* In the implicit CF, the model is evaluated by testing if it can generate a better ranked item list $\mathcal{S}_u$ for each user $u$. In the synthetic case, $\mathcal{S}_u$ contains $u$'s test items $\mathcal{G}_u$ and the rest items that are not interacted by $u$. While in the real-world case with much larger item count $|\mathcal{I}|$, we follow a common strategy [20, 24] to fix the list length $|\mathcal{S}_u|$ as 100, by randomly sampling $100-|\mathcal{G}_u|$ non-interacted items. We measure $\mathcal{S}_u$'s performance by *Recall* and *Normalized Discounted Cumulative Gain* (NDCG). Specifically, $Recall@k(u) = |\mathcal{S}_u(k) \cap \mathcal{G}_u|/|\mathcal{G}_u|$, $NDCG@k(u) = \sum_{i\in\mathcal{S}_u(k)\cap\mathcal{G}_u} 1/\log_2(p_i+1)$, where $\mathcal{S}_u(k)$ denotes truncated $\mathcal{S}_u$ at $k$ and $p_i$ denotes $i$'s rank in $\mathcal{S}_u$. Comparatively, NDCG accounts more for the position of the hit by assigning higher scores to hits at top ranks and $NDCG@1(u)=Recall@1(u)$. We choose a rather small $k$ in $\{1,3\}$, which matters more in applications [2]. The final report Recall/NCDG is the average score among all test users.

### 4.2 Synthetic Noise Experiments

Synthetic false negative instances are simulated by flipping labels of test records ($\mathcal{F}$ in Table 2). To manually inject this noise, we constantly feed a false negative into each user's memory $\mathcal{M}$ during sampling process. We control this impact by varying the size of available false negative instances in different experiments, randomly sampling $\sigma \times 100$ (%) from $\mathcal{F}$ ($\sigma \in [0,1]$). Note that $\sigma = 0$ indicates an "ideal" case where $\mathcal{M}$ is not influenced by $\mathcal{F}$. In these experiments, we fix the memory size $S_1$ as 20 (details on setup are in Appendix C.3).

#### 4.2.1 Sampling criterion

We first investigate if the high-variance based criterion in SRNS can indeed identify true negative instances which are of high-quality, by comparing with a difficulty-only strategy (*i.e.*, weight $\alpha_t = 0$). Figure 2 shows comparison results *w.r.t.* test Recall and NDCG, under different levels of noisy supervision ($\sigma$). Although increasing noisy level can harm model's performance, we can observe a consistent improvement of variance-based strategy with different $\sigma$.

#### 4.2.2 Warm-start

Motivated by [19], we propose to linearly increase the value of $\alpha_t$ as epoch number $t$ increases. Specifically, $\alpha_t = \alpha \cdot \min(t/T_0, 1)$, where $T_0$ denotes the threshold of stopping increase. In Figure 3(a)-(b), we compare this increased setting of $\alpha_t$ with another two competitor, *i.e.*, $\alpha_t = \alpha$ (flat) and $\alpha_t = \alpha \cdot \max(1 - t/T_0, 0)$ (decreased). It can be clearly observed that the increased setting of $\alpha_t$ performs better than others, as the former can better leverage variance-based criterion after false negative instances become more stable.

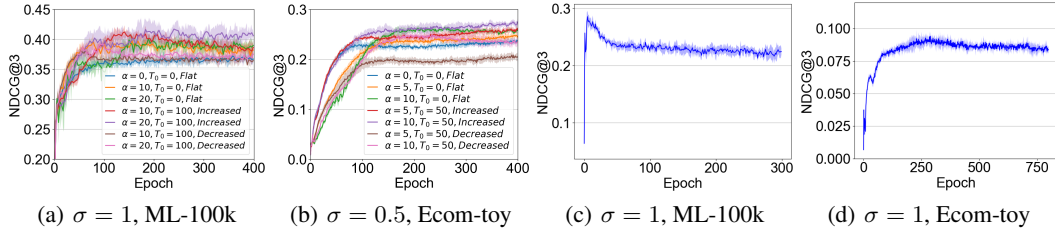

| (a) $\sigma = 1$, ML-100k | (b) $\sigma = 0.5$, Ecom-toy | (c) $\sigma = 1$, ML-100k | (d) $\sigma = 1$, Ecom-toy |

Figure 3: (a)-(b) Test NDCG vs. number of epochs on two datasets, with the error bar for STD highlighted as a shade. (c)-(d) Memorization effect of the CF model under extremely noisy supervision.

Table 3: Performance comparison *w.r.t.* test NDCG and Recall on three datasets. The last row shows relative improvement in percentage compared with the second best.

| Category | Method | Movielens-1m | | | Pinterest | | | Ecommerce | | |
| | | N@1 | N@3 | R@3 | N@1 | N@3 | R@3 | N@1 | N@3 | R@3 |
| --- | --- | --- | --- | --- | --- | --- | --- | --- | --- | --- |
| Non-sampling | ENMF | 0.1846 | 0.3021 | 0.3882 | 0.2594 | 0.4144 | 0.5284 | 0.1317 | 0.2095 | 0.2670 |
| Fixed Dist. Sampling | Uniform | 0.1744 | 0.2846 | 0.3663 | 0.2586 | 0.4136 | 0.5276 | 0.1265 | 0.2057 | 0.2640 |
| | NNCF | 0.0829 | 0.1478 | 0.1971 | 0.2292 | 0.3699 | 0.4735 | 0.0833 | 0.1420 | 0.1855 |
| Hard Negative Sampling | AOBPR | 0.1802 | 0.2905 | 0.3728 | 0.2596 | 0.4165 | 0.5319 | 0.1293 | 0.2108 | 0.2710 |
| | IRGAN | 0.1755 | 0.2877 | 0.3708 | 0.2587 | 0.4143 | 0.5282 | 0.1275 | 0.2065 | 0.2648 |
| | RNS-AS | 0.1823 | 0.2932 | 0.3754 | 0.2690 | 0.4233 | 0.5359 | 0.1335 | 0.2131 | 0.2714 |
| | AdvIR | 0.1790 | 0.2941 | 0.3792 | 0.2689 | 0.4235 | 0.5363 | 0.1357 | 0.2141 | 0.2719 |
| Proposed | SRNS | **0.1933** | **0.3070** | **0.3912** | **0.2891** | **0.4391** | **0.5486** | **0.1471** | **0.2256** | **0.2833** |
| | | 4.71% | 1.62% | 0.77% | 7.47% | 3.68% | 2.29% | 8.40% | 5.37% | 4.19% |

### 4.2.3 Bootstrapping

Finally we demonstrate SRNS's self-boosting capability, by illustrating the memorization effects of CF models in Figure 3(c)-(d). To ensure a clear observation, we inject much intenser noise during sampling process, by extending $\mathcal{F}$ to 100% and 40% of the original test set on ML-100k and Ecom-toy, respectively. Under extremely noisy supervision ($\sigma = 1$), though sampling based on difficulty only ($\alpha_t = 0$), the model's test NDCG first reaches a high level and then gradually decreases, indicating that it can avoid the impact of false negative instances at an early stage.

## 4.3 Real Data Experiments

### 4.3.1 Performance comparison

As shown in Table 3, we compare SRNS with seven baselines *w.r.t.* test NDCG and Recall on three real-world datasets. As can be seen, SRNS consistently outperforms them, achieving a relative improvement of 4.71~8.40% *w.r.t.* NDCG@1. This indicates that SRNS can sample high-quality negative instances and thus helps to learn a better CF model that ranks items more accurately. Specifically, we have following three observations. First, among all baselines, hard negative sampling approaches perform more competitively. By considering both informativeness and reliability of negative instances, our SRNS outperforms two state-of-the-art baselines, *i.e.*, RNS-AS and AdvIR, that generate hard negatives based on adversarial sampling. Second, approaches using a fixed sampling distribution perform poorly, especially NNCF that directly adopts a power distribution based on item popularity. Third, by improving sample quality, sampling-based approaches can be more effective than the non-sampling counterpart that models the whole unlabeled data. For example, ENMF performs worse than RNS-AS and AdvIR on Pinterest and Ecom.

Besides effectiveness, we also compare performance in terms of efficiency, by illustrating validation NDCG vs. wall-clock time in Figure 4(a)-(c). We observe that SRNS can converge much faster and is more stable than RNS-AS and AdvIR that use GAN based structure. For fair efficiency comparison, here we also start training SRNS from the same pretrained model as in RNS-AS and AdvIR.

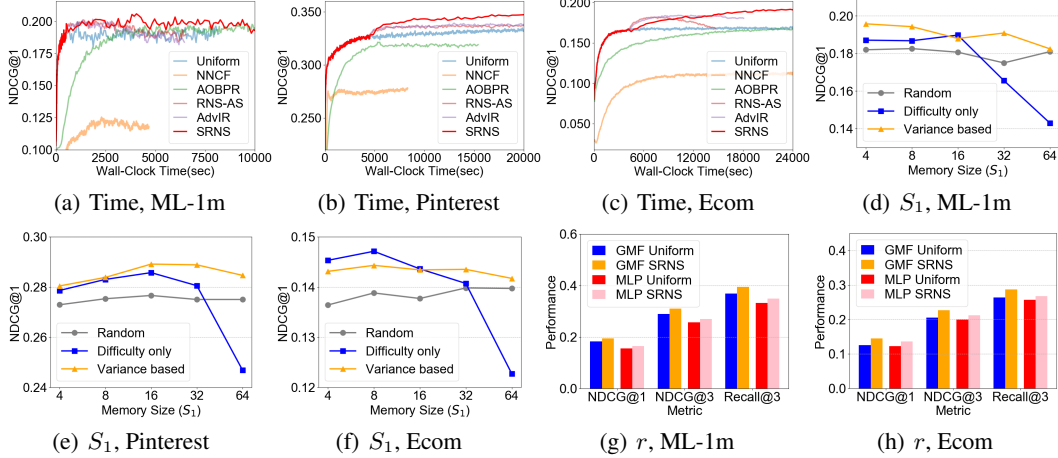

(a) Time, ML-1m    (b) Time, Pinterest    (c) Time, Ecom    (d) $S_1$, ML-1m

(e) $S_1$, Pinterest    (f) $S_1$, Ecom    (g) $r$, ML-1m    (h) $r$, Ecom

Figure 4: (a)-(c) Validation NDCG vs. wall-clock time (in seconds) on three datasets. (d)-(f) Test NDCG vs. SRNS's memory size $S_1$, using different sampling strategies on three datasets. (g)-(h) Test NDCG and Recall of Uniform and SRNS, using two $r$ (GMF and MLP), on ML-1m and Ecom.

### 4.3.2 Robustness of variance-based sampling

With the score-based updating strategy, increasing memory size $S_1$ makes SRNS more prone to the false negatives. Therefore, we further test robustness of the variance-based sampling (in (4)), by evaluating performance under different $S_1$. As illustrated in Figure 4(d)-(f), variance-based SRNS (orange line) performs stably, indicating that emphasizing high-variance can reliably obtain high-quality samples. Comparatively, difficulty-only strategy ($\alpha_t = 0$, blue line) suffers from dramatic degradation ($S_1$=32 or 64). Another strategy for avoiding false negatives is to randomly select a sample from memory [47], which performs less effectively than our approach. Note that the necessity of variance-based sampling depends on the specific real-world data and, for example, Ecom may not need this *w.r.t.* overall best NDCG@1 (Figure 4(f)). Generally, our SRNS is flexible enough to switch between different situations, by controlling importance of variance-based sampling criterion (*i.e.*, $\alpha_t$).

### 4.3.3 Varying scoring function

Finally we test SRNS's effectiveness on different $r$ including GMF and MLP [20]. As illustrated in Figure 4(g)-(h), we observe similar performance improvement of SRNS over Uniform [34] when using the above two $r$, indicating SRNS's capability of combining with different $r$. We are also interested in exploring more choices like GNN-based $r$ [44] in future study. Note that embedding dimension $F$ is set as 32 (ML-1m), 16 (Pinterest) and 8 (Ecom), respectively, as we observe similar results with different $F \in \{8, 16, 32, 64\}$ (details are in Appendix C.4).

## 5 Conclusion

In this paper, we propose a simplified and robust negative sampling approach SRNS for implicit CF, which can efficiently sample true negative instances that are of high-quality. Motivated by the empirical evidence on different negative instances, our score-based memory design and variance-based sampling criterion achieve efficiency and robustness, respectively, in negative sampling. Experimental results on both synthetic and real-world datasets demonstrate SRNS's robustness and superiorities. Finally, one interesting future works would be studying the theoretical convergence guarantee of the proposed method. We will attempt to address this issue by learning from importance sampling methods [8, 48] in stochastic optimization.

## Broader Impact

Motivated by general methodology of learning from noisy labels, this work proposes a novel negative sampling approach that aims to reliably measure the sample quality and handle false negatives correctly during sampling process. The potential risk of introducing false negatives has been overlooked by most existing negative sampling approaches in wide areas of embedding learning for text, graph, etc. This work provides a new direction for robustifying negative sampling in these areas (see details in Appendix A). Specifically, it focuses on the implicit collaborative filtering (CF) where false negatives can be a severe problem. After further combining with a simplified model design, the proposed approach achieves efficient sampling of true negative instances that are of high-quality, and can potentially benefit several downstream applications including recommender systems and user modeling.

## Acknowledgments and Disclosure of Funding

This work was supported in part by The National Key Research and Development Program of China under grant 2020AAA0106000, and the National Nature Science Foundation of China under U1936217, 61971267, 61972223, 61941117, 61861136003.

## Footnotes

[2] We also provide results on larger $k \in \{5, 10\}$ in Appendix C.4.

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
