[Supplementary Material]

# Simplify and Robustify Negative Sampling for Implicit Collaborative Filtering

**Jingtao Ding**[1][*]   **Yuhan Quan**[1]   **Quanming Yao**[1,2]   **Yong Li**[1]   **Depeng Jin**[1]
[1]Department of Electronic Engineering, Tsinghua University,
[2]4Paradigm Inc.

## A   Comparison Between Different Approaches

### A.1   General Machine Learning Approaches

Learning an implicit CF model from the positive-only data is also related to Positive-Unlabeled (PU) learning and learning from noisy labels, as the rest unobserved instances are unlabeled and noisy. Motivated by these general machine learning approaches, this paper formulates the negative sampling problem as efficient learning from unlabeled data with the presence of noisy labels, and pays more attention on those true negative instances hidden inside the massive unlabeled data. The following table and review on literatures discuss the differences between different approaches that can be adapted for this problem.

| Approaches | Learning from Positive-Unlabeled Data | Learning from Noisy Labels | Negative Sampling |
|---|---|---|---|
| Positive/Negative Class Prior | known or estimated from data [14, 31] | unknown | unknown |
| Assumption on Unlabeled Data | positive or negative labels [15, 23] | negative labels with noise [22, 32] | unobserved [27, 30, 34] |
| Handling Uncertainty of Unlabeled Data | minimizing the empirical risk estimator [13, 23] | manually designing [2, 25] or automated learning the instance weight [22, 32, 36] | sampling unobserved instances as negative labels [27, 30, 34] |

**Learning from Positive-Unlabeled Data.** Since implicit feedback data contain positive instances only, the implicit CF problem is also related to learning from positive-unlabeled (PU) data. PU learning formulates the problem as a binary classification, accounting for the fact that both positive and negative labels exist in the unlabeled data [13, 15, 23]. However, it normally requires an accurate estimation of the class-prior, which is challenging in real-world data [14, 31]. Moreover, a direct optimization on the whole unlabeled data is generally inefficient, especially for implicit CF, where an efficient training approach supporting large-scale data is necessary [23, 35]. In our proposed solution, above issues are avoided by efficiently sampling negative instances from the unlabeled data and, motivated by the idea of PU learning, we carefully distinguish those true negative instances from others.

**Learning from Noisy Labels.** By regarding unobserved instances as a combination of true negative labels and noisy labels, another choice is adapting the implicit CF into learning from noisy labels. Typical learning approaches include curriculum learning [2], self-pace learning [25] and instance re-weighting [22, 32, 36]. The first two approaches prefer easier instances during training process so as to improve robustness, while these easy instances may be ineffective for learning a CF model. Without prior information about the noisy labels, instance re-weighting approach learns the weight

---

[*]The first three authors have equal contributions.

of each instance with bi-level optimization on training and validation data [22, 32, 36]. However, the size of unlabeled data in implicit CF can approach to nearly a product of user count and item count, making above non-sampling approach become unaffordable in terms of learning efficiency. Therefore, this work focuses on negative sampling and aims to handle noisy labels correctly at the same time.

## A.2 Specific Negative Sampling Approaches

Negative sampling approaches have also been widely adopted in other domains of embedding learning for text, graph, etc. Motivated by these works that tend to leverage a simple model for capturing negative sampling distribution, we design a memory-based model that simply maintains the promising candidates with large scores. More importantly, we propose to robustify negative sampling by emphasizing high-variance samples, which is novel in both CF and other domains. The following table and review on literatures discuss the differences between different approaches.

| Domain | Text | Graph | Knowledge Graph | Collaborative Filtering |
|---|---|---|---|---|
| Learning Objective | semantic word relationships | node proximities | fact composed of head/tail entity and relation | user preferences among items |
| Vanilla Sampling Strategy | frequency-based [27] | degree-based [30, 37] | uniform [4], bernoulli [41] | uniform [34], popularity-based [10, 42] |
| Improving Sample Quality | GAN [5] | self-paced learning, GAN [16] | score-based [47], GAN [5, 7] | score-based [33, 46], GAN [12, 29] |
| Leveraging Skewness in Distribution | favouring frequent words [27] | favouring high-degree nodes [30, 37] or positive-alike nodes [43] | favouring large-scored instances [47] | none |
| Handling False Negative | none | none | none | avoiding the hardest instances [44] |

**Negative Sampling in Other Domains.** Negative sampling approaches are widely used in many tasks like word embedding [27], graph embedding [6] and knowledge graph embedding [39]. In terms of capturing the distribution of negative instances, these applications generally requires a rather simple model. For example, Word2Vec [27] sets the negative sampling distribution proportional to the 3/4 power of word frequency, which favours those frequent words. Later works on graph embedding [30, 37] readily keep this skewed distribution by adapting it to the node degree. Similarly in knowledge graph, it has been observed that negative instances with large scores are important but rare and focusing on this partial set makes the model much simpler [47]. Another recent work on negative sampling of graph representation learning proposes that the negative sampling distribution should be positively but sub-linearly correlated to their positive sampling distribution [43]. However, in terms of avoiding false negative instances, none of them have tackled this problem by designing a robust sampling approach. Since the implicit CF is a different problem where the reliability of sampled negative instances is much harder to guarantee, we propose to reduce this risk by emphasizing high variance samples. Meanwhile, motivated by above examples in other domains, we also leverage a simple model to efficiently capture the distribution of negative instances which are of high-quality.

# B Implementation Details

## B.1 Running Environment

The experiments are conducted on a single Linux server with AMD Ryzen Threadripper 2990WX@3.0GHz, 128G RAM and 4 NVIDIA GeForce RTX 2080TI-11GB. Our proposed SRNS is implemented in Tensorflow 1.14 and Python 3.7.

## B.2 Baselines

We compare the SRNS with following state-of-the-art approaches: (1) Uniform [34], which uniformly selects negative samples from the unlabeled data. (2) NNCF [10], which uses a negative sampling distribution proportional to the 3/4 power of item popularity. A hyper-parameter $s$ is the number of positive samples per item. $b$ is the number of negative samples per positive sample. (3) AOBPR [33], which improves uniform strategy by adaptively oversampling hard instances. A hyper-parameter $\lambda$ controls the skewness of distribution $\propto \exp(-\text{rk}(j|u)/\lambda)$. (4) IRGAN [38], which uses an adversarial sampler by conducting a minimax game between the recommender and the sampler. A hyper-parameter $\tau$ is the temperature in sampling distribution (Eq. (10) in [38]). (5) RNS-AS [12], which leverages adversarial sampling to generate hard negative samples. A hyper-parameter $N_s$ is size of candidate set for sampling and $\tau$ is the temperature. (6) AdvIR [29], which exploits both adversarial sampling and training (*i.e.*, adding perturbation) to generate better negative samples. $N_s$ and $\tau$ are defined similarly as above. $\epsilon$ controls the perturbation size. (7) ENMF [9], as a baseline, we also compare with an non-sampling approach that regards all the unlabeled data as negative labels and carefully assigns instance weights. A hyper-parameter $c$ controls above weight for a negative instance.

## B.3 Detail of MLP based $r$

The MLP based scoring function $r(\mathbf{p}_u, \mathbf{q}_i, \boldsymbol{\beta})$ [20] takes the concatenation of $\mathbf{p}_u$ and $\mathbf{q}_i$, *i.e.*, $\mathbf{z}_0 = [\mathbf{p}_u; \mathbf{q}_i] \in \mathbb{R}^{2F}$, as the input. Then there are $H$ hidden layers, and the $l$th layer is defined as

$$\mathbf{z}_l = \text{sigmoid}(\mathbf{W}_l \mathbf{z}_{l-1} + \mathbf{b}_l), \tag{1}$$

where $\mathbf{W}_l \in \mathbb{R}^{d_l \times d_{l-1}}$ and $\mathbf{b}_l \in \mathbb{R}^{d_l}$ denote the weight matrix and bias vector in this layer. Specifically, $d_0 = 2F$ and we set $d_l = \frac{1}{2}d_{l-1}$. The last layer outputs the prediction score $r_{ui}$, defined as

$$r_{ui} = \mathbf{W}_{H+1}^\top \mathbf{z}_H + \mathbf{b}_{H+1}, \tag{2}$$

where $\mathbf{W}_{H+1} \in \mathbb{R}^{d_H}$ and $\mathbf{b}_{H+1} \in \mathbb{R}$. The learnable parameters $\boldsymbol{\beta}$ in this MLP based $r$ are $\{\mathbf{W}_i, \mathbf{b}_i\}(i = 1, ..., H + 1)$.

## B.4 Hyper-parameter Tuning

Our SRNS's hyper-parameters can be divided into three parts: (1) sampling related part, including memory size $S_1$, expansion size $S_2$, temperature $\tau$, variance-based criterion weight $\alpha$, warm-start epoch number $T_0$. (2) $r$ related part, including embedding dimension $F$ and hidden layer number $H$. (3) optimization related part, including learning rate $lr$ and L2 regularization $reg$.

In synthetic noise experiments, since we do not explicitly split a validation set on synthetic data, we draw two different train/test splits. The hyper-parameters are searched in the first round and afterwards are kept constant in another round. Note that the false negative instances ($\mathcal{F}$) in there two rounds are also independent with each other, as they are simulated by random sampling from the corresponding test set. We run each synthetic data experiment 400 epochs without early stopping and repeat five times. The scoring function $r$ is GMF [20]. The memory size $S_1/S_2$ are fixed as 20/20. The temperature $\tau$ is 1. Adam optimizer with $\beta_1 = 0.9$, $\beta_2 = 0.999$ is used and the mini-batch size is set to 1024. The lazy-update epoch number $E = 1$. The rest hyper-parameters are tuned according to average NDCG@3 in the last 50 training epochs. Specifically, first we use grid search to find the best group of non-sampling related hyper-parameters, *i.e.*, $(F, lr, reg)$, using the vanilla Uniform method [34] as the negative sampling strategy. Then we fix $(F, lr, reg)$ and search the rest sampling related hyper-parameters, *i.e.*, $(\alpha, T_0)$, under different settings of noisy supervision ($\sigma$). See Table 4 for detailed information.

In real data experiments, we conduct the standard procedure to split train/validation/test set. We run 400 epochs and terminate training if validation performance does not improve for 100 epochs, which has also been repeated five times. Both GMF and MLP (defined in Appendix B.3) are tested. Adam optimizer with $\beta_1 = 0.9$, $\beta_2 = 0.999$ is used and the mini-batch size is set to 1024. The lazy-update epoch number $E = 1$. The embedding dimension $F$ is set as 32 (ML-1m), 16 (Pinterest) and 8 (Ecom), respectively. We further show similar results with different $F \in \{8, 16, 32, 64\}$ in Appendix C.4. The rest hyper-parameters are tuned according to the best NDCG@1 on the validation set. Specifically, first we use grid search to find the best group of non-sampling related

Table 4: SRNS's hyper-parameter exploration in synthetic noise experiments (Section 4.2)

| Hyper-parameter | Tuning Range | Opt. (Ecom-toy) | Opt. (ML100k) |
|---|---|---|---|
| $lr$ | $[5, 10, 50] \times 10^{-4}$ | 0.001 | 0.001 |
| $reg$ | $[0, 1, 10] \times 10^{-3}$ | 0.0 | 0.001 |
| $F$ | $[8, 16, 32]$ | 32 | 8 |
| $\alpha$ | $[5.0, 10.0, 20.0, 50.0]$ | - | - |
| $T_0$ | $[50, 100]$ | - | - |

hyper-parameters, *i.e.*, $(lr, reg)$, using the vanilla Uniform method [34] as the negative sampling strategy (For MLP based $r$ we also search $H$). Then we fix them and search the rest sampling related hyper-parameters, *i.e.*, $(\tau, \alpha, T_0, S_1, S_2/S_1)$. To ease the tuning process, we first fix $\alpha$ and $T_0$ as 0 (difficulty-only sampling), then search the best $(\tau, S_1, S_2/S_1)$. After that we fix them and search the best group of $(\alpha, T_0)$ (variance-based sampling). Also, we repeat above step by changing memory size $S_1$ to its next or previous value. For example, if current best $S_1$ is 16, we further test 8 and 32. Note that we do not search sampling related hyper-parameters when using MLP based $r$, by directly using those for GMF. See Table 5 for detailed information.

Table 5: SRNS's hyper-parameter exploration in real data experiments (Section 4.3)

| | Para. | Tuning Range | Opt. (Ecom) | Opt. (ML1m) | Opt. (Pinterest) |
|---|---|---|---|---|---|
| SRNS GMF | $lr$ | $[5, 10, 50, 100] \times 10^{-4}$ | 0.001 | 0.001 | 0.001 |
| | $reg$ | $[0, 1, 10, 100] \times 10^{-4}$ | 0.001 | 0.01 | 0.0 |
| | $\tau$ | $[0.5, 1.0, 2.0, 10.0]$ | 2.0 | 10.0 | 10.0 |
| | $\alpha$ | $[0.1, 1.0, 2.0, 5.0, 10.0, 20.0, 50.0]$ | 0.0 | 5.0 | 5.0 |
| | $T_0$ | $[25, 50, 100]$ | 25 | 50 | 50 |
| | $S_1$ | $[2, 4, 8, 16, 32]$ | 8 | 8 | 16 |
| | $S_2/S_1$ | $[1, 2, 4, 8]$ | 2 | 8 | 4 |
| SRNS MLP | $lr$ | $[5, 10, 50] \times 10^{-4}$ | 0.001 | 0.001 | - |
| | $reg$ | $[0, 1, 10, 100] \times 10^{-4}$ | 0.001 | 0.01 | - |
| | $H$ | $[0, 1, 2, 3]$ | 3 | 3 | - |

As for the baselines listed in Appendix B.2, except Uniform [34] that have been tuned as above, others have also been carefully tuned according to their validation NDCG@1. For IRGAN, RNS-AS and AdvIR using GAN-based structure, we use a pretrained model (*i.e.*, trained under Uniform) to initialize. See Table 6 for detailed information.

## B.5   Evaluation Metrics

As defined in Section 4.1, our used metrics, *i.e.*, Recall and NDCG, can provide a comprehensive evaluation of model performance. The former measures whether the ground truth item is presented on the ranked list, while the latter measures the performance at a finer granularity by accounting for the position of hit. The two datasets (ML-100k and Ecom-toy) used in synthetic noise experiments are rather small, with the item count $|\mathcal{I}|$ between 1000~2000, while the rest three in real data experiments are much larger, with the highest value as 59290 (Ecom). Thus in real data experiments, we follow a common strategy [20, 24] to fix the list length $|\mathcal{S}_u|$ as 100, by randomly sampling $100 - |\mathcal{G}_u|$ non-interacted items, because ranking the whole item set for each user is too time-consuming during evaluation. When reporting NDCG@k and Recall@k, we choose a rather small value of truncated length $k \in \{1, 3\}$, because of following two reasons: (1) In real applications of implicit CF like recommender systems, users tend to browse the items at first few positions of a list, making the accuracy of rest recommended items less important. (2) In real data experiments we fix the length of a ranked list as 100, thus choosing a large $k$ may make this task too easy.

## B.6   Variance Computation

To calculate the prediction variance $\text{std}[P_{\text{pos}}(k|u, i)]$ (Eq. (4)) of each candidate instance $(u, k)$ stored in the memory $\mathcal{M}_u$, we directly use the prediction results from previous iterations, without any extra

Table 6: Baselines' hyper-parameter exploration in real data experiments (Section 4.3)

| Method | Para. | Tuning Range | Opt. (Ecom) | Opt. (ML1m) | Opt. (Pinterest) |
|--------|-------|--------------|-------------|-------------|------------------|
| Uniform | $lr$ | $[5, 10, 50, 100] \times 10^{-4}$ | 0.001 | 0.001 | 0.001 |
| | $reg$ | $[0, 1, 10, 100] \times 10^{-4}$ | 0.001 | 0.01 | 0.0 |
| NNCF | $lr$ | $[5, 10, 50, 100] \times 10^{-4}$ | 0.001 | 0.001 | 0.001 |
| | $reg$ | $[0, 1, 10, 100] \times 10^{-4}$ | 0.0 | 0.0 | 0.0 |
| | $b$ | $[32, 64, 128, 256, 512, 1024, 2048]$ | 32 | 2048 | 2048 |
| | $s$ | $[1, 2, 4]$ | 2 | 2 | 2 |
| ENMF | $lr$ | $[5, 10, 50, 100] \times 10^{-4}$ | 0.01 | 0.01 | 0.005 |
| | $reg$ | $[0, 1, 10, 100] \times 10^{-4}$ | 0.001 | 0.0001 | 0.0 |
| | $c$ | $[0.01, 0.03, 0.05, 0.07, 0.1, 0.3, 0.5, 0.7]$ | 0.1 | 0.3 | 0.01 |
| AOBPR | $lr$ | $[5, 10, 50, 100] \times 10^{-4}$ | 0.0005 | 0.0005 | 0.0005 |
| | $reg$ | $[0, 1, 10, 100] \times 10^{-4}$ | 0.001 | 0.01 | 0.0 |
| | $\lambda$ | $[5, 10, 20, 50, 100, 200, 500, 1000, 2000]$ | 10 | 1000 | 2000 |
| IRGAN | $lr$ | $[5, 10, 50, 100] \times 10^{-4}$ | 0.0005 | 0.0005 | 0.0005 |
| | $reg$ | $[0, 1, 10, 100] \times 10^{-4}$ | 0.001 | 0.001 | 0.001 |
| | $\tau$ | $[0.5, 1.0, 2.0]$ | 2.0 | 1.0 | 1.0 |
| RNS-AS | $lr$ | $[5, 10, 50, 100] \times 10^{-4}$ | 0.001 | 0.0005 | 0.0005 |
| | $reg$ | $[0, 1, 10, 100] \times 10^{-4}$ | 0.0 | 0.001 | 0.01 |
| | $\tau$ | $[0.5, 1.0, 2.0, 10.0]$ | 1.0 | 0.5 | 0.5 |
| | $N_s$ | $[10, 20, 30, 40]$ | 10 | 10 | 10 |
| AdvIR | $lr$ | $[5, 10, 50, 100] \times 10^{-4}$ | 0.0005 | 0.0005 | 0.0005 |
| | $reg$ | $[0, 1, 10, 100] \times 10^{-4}$ | 0.0 | 0.0001 | 0.001 |
| | $\epsilon$ | $[1, 10, 100] \times 10^{-2}$ | 0.01 | 0.01 | 0.01 |
| | $\tau$ | $[0.5, 1.0, 2.0, 10.0]$ | 1.0 | 1.0 | 1.0 |
| | $N_s$ | $[10, 20, 30, 40]$ | 10 | 10 | 10 |

forward or backward passes in the $r$. In our experiments, we determine the uncertainty only based on the latest 5 epochs (Eq. (5)). In our implementation, we consider the prediction probability in the latest few epochs, which is due to following two reasons: (1) prediction history near the beginning period of training process is not stable for all kinds of negative instances, and thus can be excluded from the computation. (2) this implementation makes the overhead constant ($O(1)$) for each sampling operation.

In real data experiments where the datasets are much larger, it is time-consuming to compute the prediction probability ($P_{\text{pos}}$) for all user-item pairs ($|\mathcal{U}| \cdot |\mathcal{I}|$) at each epoch. Thus we prune the item space for each user's memory update process, so as to avoid logging $P_{\text{pos}}$ for all items. Specifically, for $u$ at $t$th training epoch, the newly extended candidates in $\bar{\mathcal{M}}_u$ can only be randomly sampled from an item set, denoted as $var\_set_u$, which has already been generated at $(t-5)$th epoch. At the mean time, for $v \in var\_set_u$, we log $P_{\text{pos}}(v|u, i)$ values at the subsequent 5 epochs. Therefore, among $u$'s memory $\mathcal{M}_u$, besides the original items that have been maintained from previous epochs, the newly added items also have the $P_{\text{pos}}$ history in the latest 5 epochs, which supports the variance computation above. In terms of the time complexity for tracking above scores, it is about $O(|var\_set_u||\mathcal{U}|T)$ without backward computation, which should be with the same magnitude as required in sampling, i.e., $O(|\mathcal{R}|(S_1 + S_2)T/E)$ for one epoch. As for the memory cost, the extra space with a complexity of $O(|\mathcal{U}||var\_set_u|)$ is used for storing prediction scores, which is affordable in our experiments on a single-machine. Note that $var\_set_u$ is also generated by random sampling from $u$'s non-interacted items, and its size is larger than memory size $S_1$, but much smaller than item count $|\mathcal{I}|$. Currently we do not consider $|var\_set_u|$ as a hyper-parameters for tuning, and choose a default value instead, i.e., 3000 (ML-1m) and 600 (Pinterest, Ecom), respectively. We can further tune it to achieve the balance between accuracy and scalability. Moreover, the size of $var\_set_u$ can be reduced in a larger dataset. Previous work in recommender system has observed that the candidate space for negative sampling can be largely reduced by assigning each user a different set of candidates before the training [11]. For a larger dataset containing more users and items, it may be much sparser and the size of $var\_set_u$ can also be reduced.

# C   Experiment Details

## C.1   Dataset Description

We choose following four raw datasets and build five datasets for performance evaluation.

- **Movielens (ML)-100k**[2]. This is a widely used movie-rating dataset containing 100,000 ratings on movies from 1 to 5. We follow the common preprocessing to convert it into implicit feedback data, regarding those high-rated records $(4 \sim 5)$ as positive labels [29, 38].
- **Movielens (ML)-1m**[3]. Similarly to ML-100k, this large dataset contains 1,000,000 ratings. After similar converting procedure, we filter out users with less than 5 records.
- **Pinterest**[4]. This implicit feedback dataset is constructed by [17] for a task of image recommendation, and has been used for evaluating the implicit CF task [20].
- **Ecommerce (Ecom)**. This implicit feedback dataset is a subset of users' item-click records in a real-word E-commerce website between 2017/06 and 2017/07. For data preprocessing, we filter out users/items with less than 4 records, so as to overcome the problem of high sparsity. After that, we further obtain a toy dataset, denoted as **Ecom-toy**, by retaining top 1,000 users and 2,000 items sorted by number of records.

## C.2   Details of Figure 1

The experiment is conducted on ML-100k dataset, using the same train/test split as synthetic noise experiments. We use GMF as the $r$ and Uniform [34] as the negative sampling strategy. By flipping labels of groundtruth records in the test set, we are able to obtain a set of false negative instances (FN) that are in fact positive labeled but unobserved during the negative sampling process. Besides uniformly sampling negative instances (UN) to update the model, we simultaneously obtain a series of hard negative instances (HN) with different difficulty $D$. In following analysis, we adopt a simple yet effective strategy to control $D$ of a obtained HN: 1) uniformly sample $D$ candidates from $\{(u, j)|j \notin \mathcal{R}_u\}$; 2) select the negative instance with the highest value of $r_{uj}$. When $D$ gets higher, HN becomes much harder. UN is the same as HN with $D = 1$.

As in Figure 1(a), we have a closer look at the negative instances' distribution in terms of their positive-label probabilities $P_{\text{pos}}$ that are proportional to the prediction scores. This is motivated by [47] that has observed a skewed distribution of negative instances when learning knowledge graph embeddings. Specifically, (a) is the distribution of negative instances $\{(u, j)|u \in \mathcal{U}, j \notin \mathcal{R}_u\}$ at 5 timestamps. We measure the *complementary cumulative distribution function* (CCDF) $F(x) = P(P_{\text{pos}} \geq x)$ to show the proportion of negative instances that satisfy $P_{\text{pos}} \geq x$. Since hard negative instances generally have large $P_{\text{pos}}$, we compare them with those false negative instances *w.r.t.* $P_{\text{pos}}$ (Figure 1(b)). We use the median value ($p50$) to represent each set. Then in Figure 1(c), we further analyze the possibility of using $P_{\text{pos}}$ to discriminate above two sets of negative instances. Specifically, under different hard negative sampling strategies, we calculate the *label error ratio* in each mini-batch, *i.e.*, $LER = {}^{(\#\ of\ false\ negative\ samples)}/_{(\#\ of\ all\ selected\ negative\ samples)}$. Unlike others, false negative instances follow the similar distribution as those positive instances in training data. Thus the model can ideally become more and more confident about predicting them as positive instances, and the corresponding variance of $P_{\text{pos}}$ is low. Finally, to validate this, we compare $P_{\text{pos}}$'s variance between different types of negative instances in Figure 1(d). The normalized variance is measured by the ratio between standard deviation and mean value.

## C.3   Synthetic Noise Experiments

To control the impact of false negative instances on the sampling process, we manually inject noisy labels by slightly modifying each user's memory $\mathcal{M}$ that stores $S_1$ candidate negative instances. Specifically, for user $u$, there is always an instance in $\mathcal{M}_u$ that is randomly sampled from $u$'s false negative set $\mathcal{F}_u$, and this instance is also dynamically updated together with $\mathcal{M}_u$. As for the rest $S_1 - 1$ candidates in $\mathcal{M}_u$, they cannot be selected from $\mathcal{F}_u$. To control the noise ratio, we vary

**(a)** $\sigma = 0$, ML-100k    **(b)** $\sigma = 0.2$, ML-100k    **(c)** $\sigma = 0.6$, ML-100k    **(d)** $\sigma = 0.8$, ML-100k

**(e)** $\sigma = 0$, Ecom-toy    **(f)** $\sigma = 0.3$, Ecom-toy    **(g)** $\sigma = 0.7$, Ecom-toy    **(h)** $\sigma = 1$, Ecom-toy

Figure 5: Detailed results of Figure 3: Test NDCG vs. number of epochs on two datasets, with the error bar for STD highlighted as a shade.

Table 7: Detailed investigation of "warm-start" on ML-100k, $\sigma = 1.0$ (Figure 3(a)).

|  | $T_0/\alpha$ | 5 | 10 | 20 | 50 |
|---|---|---|---|---|---|
| Flat | 0 | 0.3703±0.0033 | 0.3811±0.0048 | 0.3876±0.0054 | 0.4004±0.0112 |
| Increased | 50 | 0.3734±0.0045 | 0.3931±0.0097 | 0.3924±0.0050 | 0.3965±0.0099 |
|  | 100 | 0.3725±0.0111 | 0.3850±0.0075 | **0.4062±0.0073** | 0.3844±0.0078 |
| Decreased | 50 | 0.3631±0.0066 | 0.3677±0.0049 | 0.3700±0.0064 | 0.3623±0.0108 |
|  | 100 | 0.3620±0.0063 | 0.3650±0.0039 | 0.3710±0.0062 | 0.3719±0.0055 |

the size of false negative set by randomly sampling $\sigma \times 100$ (%) from $\mathcal{F}_u$ ($\sigma \in [0,1]$). Note that $\sigma = 0$ indicates an "ideal" case where $\mathcal{M}_u$ is not influenced by $\mathcal{F}_u$. In these experiments, we fix the memory size $S_1$ as 20.

Note that in the "ideal" case with no explicit noise, SRNS still largely outperforms in Ecom-toy dataset, which is also reasonable given the fact that $\mathcal{F}$ cannot ideally cover all the false negative instances hidden in unlabeled data.

In each figure of Figure 5, the blue curve represents the result of difficulty-only sampling strategy, while the grey curve and orange curve both represent those of the SRNS, with the difference on whether to linearly increase weight $\alpha_t$ during training process. It can be clearly observed that the "warm-start" setting of $\alpha_t$ performs better than a fixed-value setting, as the former better leverages prediction variance after false negative instances become stable. More detailed investigation on different settings of $\alpha_t$ are shown in following two tables.

Table 8: Detailed investigation of "warm-start" on Ecom-toy, $\sigma = 0.5$ (Figure 3(b)).

|  | $T_0/\alpha$ | 5 | 10 | 20 | 50 |
|---|---|---|---|---|---|
| Flat | 0 | 0.2449±0.0052 | 0.2557±0.010 | 0.2525±0.0063 | 0.2343±0.0019 |
| Increased | 50 | 0.2574±0.0051 | 0.2702±0.0048 | 0.2515±0.0053 | 0.2329±0.0090 |
|  | 100 | 0.2464±0.0051 | **0.2581±0.0072** | 0.2636±0.0091 | 0.2267±0.0092 |
| Decreased | 50 | 0.2037±0.0064 | 0.2351±0.0081 | 0.2367±0.0091 | 0.2365±0.0110 |
|  | 100 | 0.2120±0.0029 | 0.2351±0.0062 | 0.2348±0.0051 | 0.2513±0.0053 |

| (a) $F$, N@1, Pinterest | (b) $F$, N@3, Pinterest | (c) $F$, R@3, Pinterest |
| (d) $F$, N@1, Ecom | (e) $F$, N@3, Ecom | (f) $F$, R@3, Ecom |

Figure 6: Varying embedding dimension $F$: Test NDCG/Recall of Uniform and SRNS approaches, using different embedding size $F$, on Pinterest and Ecom, respectively.

## C.4 Real Data Experiments

### C.4.1 Performance on different embedding size

Figure 6 shows test NDCG of Uniform and SRNS approaches using different embedding size $F$. The scoring function $r$ is GMF. Again we can observe consistent improvement of SRNS over Uniform when $F \in \{8, 16, 32, 64\}$. Although increasing $F$ should have improved performance, we observe instead that $F = 16$ performs the best on Pinterest dataset, which conforms to a previous work (Figure 4 in [20]).

### C.4.2 Supplementary results on different evaluation metrics

We list the rest results ($k = 5/10$) in the following Table 9. It can be observed that the proposed SRNS still outperforms various baselines.

Table 9: Performance comparison *w.r.t.* longer recommendation list length $k$.

| Method | Movielens-1m | | | | Pinterest | | | |
| --- | --- | --- | --- | --- | --- | --- | --- | --- |
| | N@5 | N@10 | R@5 | R@10 | N@5 | N@10 | R@5 | R@10 |
| ENMF | 0.3507 | 0.4030 | **0.5066** | 0.6682 | 0.4777 | 0.5370 | 0.6824 | **0.8643** |
| Uniform | 0.3348 | 0.3932 | 0.4884 | 0.6689 | 0.4750 | 0.5323 | 0.6766 | 0.8524 |
| NNCF | 0.1835 | 0.2302 | 0.2840 | 0.4297 | 0.4309 | 0.4925 | 0.6218 | 0.8114 |
| AOBPR | 0.3428 | 0.4005 | 0.5002 | **0.6780** | 0.4790 | 0.5375 | 0.6837 | 0.8631 |
| IRGAN | 0.3372 | 0.3957 | 0.4912 | 0.6714 | 0.4750 | 0.5327 | 0.6758 | 0.8528 |
| RNS-AS | 0.3443 | 0.3993 | 0.4992 | 0.6684 | 0.4839 | 0.5390 | 0.6832 | 0.8523 |
| AdvIR | 0.3445 | 0.3973 | 0.5018 | 0.6644 | 0.4843 | 0.5393 | 0.6839 | 0.8527 |
| SRNS | **0.3527** | **0.4093** | 0.5025 | 0.6712 | **0.4971** | **0.5505** | **0.6894** | 0.8531 |
| | 0.57% | 1.56% | -0.81% | -1.00% | 2.64% | 2.08% | 0.80% | -1.30% |

Figure 7 shows supplementary results, *w.r.t.* NDCG@3 and Recall@3, of Figure 4(d)-(f), which are similar to those findings *w.r.t.* NDCG@1.

### C.4.3 Supplementary results on different experimental settings

Note that the literatures have two major experimental settings when transforming explicit feedbacks into implicit ones, *i.e.*, using only 4/5 star ratings as positive or using all ratings as positive, while our paper adopts the first one when handling **Movielens** dataset. Here we also provide results for the second setting on **Movielens** dataset. According to Table 10, both SRNS and RNS-AS achieve

(a) $S_1$, N@3, ML-1m     (b) $S_1$, R@3, ML-1m     (c) $S_1$, N@3, Pinterest

(d) $S_1$,R@3, Pinterest     (e) $S_1$, N@3, Ecom     (f) $S_1$, R@3, Ecom

Figure 7: Detailed results of Figure 4(e) and (f): Test NDCG@3/Recall@3 vs. SRNS's memory size $S_1$, using different sampling strategies on three datasets.

the best performance, while the latter only adopts the difficulty-only strategy for sampling negatives. Therefore, the noise of false negative instances has less impact under this setting. However, our proposed SRNS is flexible enough to include difficulty-only strategy as a special case.

Table 10: Performance comparison of using all ratings as positive on Movislens-1m

| Method | Movielens-1m | | | | |
|---|---|---|---|---|---|
| | N@1 | N@3 | N@5 | R@3 | R@5 |
| ENMF | 0.1891 | 0.3028 | 0.3538 | 0.3866 | **0.5108** |
| Uniform | 0.1762 | 0.2875 | 0.3375 | 0.3705 | 0.4924 |
| NNCF | 0.1086 | 0.1805 | 0.2217 | 0.2339 | 0.3346 |
| AOBPR | 0.1800 | 0.2904 | 0.3402 | 0.3727 | 0.4935 |
| IRGAN | 0.1745 | 0.2870 | 0.3357 | 0.3712 | 0.4894 |
| RNS-AS | **0.1927** | 0.3075 | **0.3547** | **0.3922** | 0.5070 |
| AdvIR | 0.1901 | 0.3013 | 0.3531 | 0.3843 | 0.5101 |
| SRNS | **0.1927** | **0.3077** | 0.3542 | **0.3922** | 0.5055 |
| | 0.00% | 0.07% | -0.14% | 0.00% | -1.04% |

## Footnotes

[2]https://grouplens.org/datasets/movielens/100k

[3]https://grouplens.org/datasets/movielens/1m

[4]https://pinterest.com