[Reviews · NeurIPS 2020]

Review 1

Summary and Contributions: The paper proposes a negative sampling strategy for mining hard negatives in the implicit positive-only collaborative filtering setting. Authors first make an observation that false negatives have a lower score variance throughout training. They then incorporate the variance into the negative sampling approach by combining it with model scores. The sampling thus prioritises negatives that have both high model scores and high variance making them less likely to be false negatives. Experiments on synthetic and real datasets show gains from applying this strategy.

Strengths: I think the problem of eliminating or reducing false negatives in implicit collaborative filtering setting is important. As training progresses and model gets better, sampling according to high scores has increasingly higher chance of including false negatives and hurting model accuracy. Authors make an interesting observation regarding the score variance over training updates and its relation to sample quality. I think this opens up avenues for further exploration in this area and can lead to better samplers.

Weaknesses: I don't think the proposed algorithm to leverage the variance is fully sound. The user memory set Mu is updated on each iteration by first uniformly sampling additional items and then retaining those with higher scores (with probability proportional to softmax of the score). First, for datasets that have lots of items, uniform sampling is very unlikely to produce hard negatives with high scores so this procedure can be highly inefficient. Second, since new samples are likely to have lower scores, one either has to increase the temperature or leave Mu relatively static between iterations. If Mu is static then training can saturate and the model can overfit to these negative examples. And if Mu is frequently changed under high temperature, then I don't see how the std can be accurately estimated in Equation 4. To get a reliable estimate each item has to stay in Mu for a number of updates and initial estimates can be highly inaccurate. Finally, to estimate std one has to repeatedly re-compute all relevant pairwise probabilities each time user or item embedding is changed. This can be very expensive since each model update can trigger many such re-calculations. I think all these issues and trade-offs need to be addresses and empirically validated before this work can be considered for publication. ***update After reading author rebuttal I still have concerns about the complexity of the proposed sampling approach. From appendix B.6 var_set_u has between 600 and 3K items for each user that are tracked for 5 iterations. This means that for the Ecommerce dataset close to 200M user-item scores are tracked and it quickly explodes from there with over a billion scores for the Netflix dataset for example. Since this is one of the key parts of the algorithm I think the memory overhead needs to be addressed. Specifically, I think that larger larger scale experiments are required that directly show how this overhead is handled, otherwise this approach has very limited practical applicability.

Correctness: Further trade-off and complexity analysis is need for the proposed algorithm.

Clarity: Paper is easy to follow but has grammatical errors.

Relation to Prior Work: Prior work and contribution are clearly discussed.

Reproducibility: Yes

Additional Feedback:


Review 2

Summary and Contributions: The paper addresses the problem of item recommendation in recommender systems, esp. the sampling process for potential negative items, if only positive items are observed. The authors propose a new negative item sampling criterion, a linear combination of the current predicted score and the variance of this scores over the last learning iterations. In experiments on 3 datasets they show that their method outperforms several state-of-the-art methods using different sampling strategies consistently by 0.77-8.4%.

Strengths: - the proposed solution is very simple. - the related work is nicely and systematically described. - extensive experiments have been conducted.

Weaknesses: - experimental results cannot be compared directly with published results due to different experimental conditions.

Correctness: Yes.

Clarity: Yes, the paper is mostly well written. Some minor grammar issues should be fixed.

Relation to Prior Work: Yes, related work is nicely and systematically described.

Reproducibility: Yes

Additional Feedback: The evaluation has been conducted for very short recommendation lists (@1, @3), while the literature often also evaluates longer lists (@5, @10, @20 and beyond). Also covering some longer lists would add another aspect to the evaluation. Experimental results cannot be compared directly with published results due to different experimental conditions. For example, IRGAN [37] evaluates only on MovieLens100k, ENMF [9] labeles all rated items as positive, not just the 4-5 rated ones, and evaluates on way higher recommendation list lengths (50,100, 200). Also covering the experimental conditions of some major baselines and reporting in the appendix would make methods easier to compare. --- response to the authors' response thank you for the additional experimental results. as the literature has two major experimental settings (using only 4+5 star ratings as positive or using all ratings a positive), it might be worthwhile to provide results for both in the final version of the paper (maybe in an appendix), making it easy for readers to compare against all published results.


Review 3

Summary and Contributions: This paper proposed an advanced negative sampling approach based on authors analysis on the property of true negative samples (low variance during training). Based on such analysis, the paper propose to combine two tricks to explore such property: score-based memory update and variance-based negative sampling. For score-based memory update, the paper suggests to maintain a small group of items for each user as negative sampling pool. The pool is optimized to contain "hard negative samples" with temperature scaled sampling. For variance-based sampling, the paper suggests to put standard deviation into negative example selection criterion. Experiments shows the proposed method, SRNS, outperforms multiple recently proposed models in terms of NDCG@1 and 3, Recall@1 and 3.

Strengths: The paper is well written with reasonable motivation. An efficient and effective negative sampling problem is interesting.

Weaknesses: Some concepts are used without explanation. For example "hard negative sample", while the authors refer this term to paper [28], I cannot find any such description in the referred paper and really confused about what is a so-called hard negative sample. It took me a while to believe that the negative samples are those that receive a high probability of likes by a given model. The analysis is relatively weak as it only uses a synthetic data (Movielens-100k) to convince the reader that the observation the authors found (true negative sample shows lower confidence score variance during training) is generalizable. It is hard to justify without repeated observations on other datasets. Experiments does not fully reflect performance of the trained model in recommendation tasks since the NDCG and Recall reported is only for 1 and 3.

Correctness: I cannot tell since the low variance of true negative is only demonstrated through a synthetic data. Need further justification. Only if the concern is solved, the rest of the paper claimed is valid.

Clarity: In general, yes. But please define the "hard negative sample".

Relation to Prior Work: Yes, the authors provide one big section on describing the existing negative sampling approaches.

Reproducibility: Yes

Additional Feedback: The proposed solution while not seen in recommendation literature, they are somewhat looks familiar in related fields. 1) maintain small sample pool common to see in other closely related fields (experience replay in Continuous learning, proposing samples for training Out-of-Distribution detection model). Maybe it is better to discuss a bit on the difference. I appreciate the authors feedback on my questions. After reading author feedback and reviewer's discussion, I would maintain my review score due to: 1. the scalability of the proposed approach seems a concern. This should be addressed. 2. "low variance of true negative" claim is not convincing. The experiment should show it instead of refering to active learning literature. they are different.


Review 4

Summary and Contributions: This paper proposes a simple and robust negative sampling method. First, it analyzes the distribution of false-negative instances, and distinguish between harder negative samples and false-negative samples by using the prediction variance of negative samples. Experimental results show that the proposed variance-based negative sampling can improve existing difficulty-based negative sampling.

Strengths: 1. This paper effectively analyzes the differences between hard negative samples and false negative samples. 2. This paper is well-written and easy to follow. Also, the time complexity is analyzed well. 3. It shows a thorough experimental evaluation, including various ablation studies and extensive evaluation for synthetic and real-world datasets.

Weaknesses: 1. Although the proposed methods show fast convergence time in terms of efficiency, it is unclear whether the proposed one is much faster than existing sampling methods. 2. For various methods for GMF and MLP (in Figures 4(g) and 4(h)), it would be better to compare the proposed model with state-of-the-art sampling methods such as RNS-AS and AdvIR, instead of Uniform. 3. The performance gain seems to be marginal as the number of recommended items increases. In Table 3, the performance gain of NDCG@3 is less significant than that of NDCG@1. 4. Candidate-based sampling can also be another baseline method. Also, it needs to consider another reinforced-based sampling method. Please refer to the following references. - Jingtao Ding et al., "An Improved Sampler for Bayesian Personalized Ranking by Leveraging View Data," WWW 2018 http://staff.ustc.edu.cn/~hexn/papers/www18-improvedBPR.pdf - Jingtao Ding et al., "Reinforced Negative Sampling for Recommendation with Exposure Data," IJCAI 2019 https://www.ijcai.org/Proceedings/2019/0309.pdf

Correctness: The proposed method is correct. However, it would be better to show a more thorough evaluation, including additional baseline methods.

Clarity: This paper is well-written, and the organization of this paper is clear. Minor opinion: Simplify -> Simple and Robustify -> Robust seem to be more natural.

Relation to Prior Work: It would welcome to include several sampling methods, as mentioned in weak points.

Reproducibility: Yes

Additional Feedback:

[Author Response · NeurIPS 2020]



To REVIEWER 1: *Q1*. *Uniform sampling is very unlikely to produce hard negatives with high scores.* **Reply.** Yes, this
is also why uniform sampling is not able to generate high quality negative samples. Thus, previous works (IRGAN [37],
AdvIR [28]) tried to fit the real negative sampling distribution with techniques of adversarial learning. However, by
emphasizing hard negative samples with large scores, they overlook the risk of introducing false negative instances.
To solve this problem, we propose to robustify negative sampling by favouring high-variance samples. Moreover, we
simultaneously design a simplified memory-based solution for efficient sampling.

*Q2*. *Since new samples are likely to have lower scores, one either has to increase the temperature or leave Mu relatively*
*static between iterations.* **Reply.** Since CF model can memorize easy training instances first and gradually adapt to
hard instances, *a.k.a.* memorization effect [1] (See our experiment results in Fig. 3c/d.), it is unnecessary to avoid
introducing new samples into negative sampling. After several training epochs, model is well trained and even new
samples can have high scores.

*Q3*. *how the std can be accurately estimated in Equation 4? And estimating std is expansive.* **Reply.** Please check
Appendix B.6 for details. For each candidate sample stored in memory Mu, we directly use its corresponding prediction
probability in the latest 5 epochs to compute the std. These prediction results have already been logged even if this
sample has just entered Mu. Without any extra forward or backward passes, the computation overhead is constant $(O(1))$
for each sampling operation.

To REVIEWER 2: *Q1*. *Evaluation results on longer lists (@5, @10, @20 and beyond).*

**Reply.** In real applications, it is more important to rank the suitable items at top positions of a list. Therefore, a smaller value of $K$ in evaluation emphasizes more on this capability. Previous works [19,23] set $K$ as 1$\sim$10 (out of 100 evaluated items) and 20 (out of 2000 items), respectively. Due to space limitation, we only report the results at $K = 1/3$. As suggested by reviewers, we list the rest results ($K = 5/10$) in the following table. It can be observed that the proposed SRNS still outperforms various baselines.

| Method | Movielens-1m | | | | Pinterest | | | |
|---|---|---|---|---|---|---|---|---|
| | N@5 | N@10 | R@5 | R@10 | N@5 | N@10 | R@5 | R@10 |
| ENMF | 0.3507 | 0.4030 | **0.5066** | 0.6682 | 0.4777 | 0.5370 | 0.6824 | **0.8643** |
| Uniform | 0.3348 | 0.3932 | 0.4884 | 0.6689 | 0.4750 | 0.5323 | 0.6766 | 0.8524 |
| NNCF | 0.1835 | 0.2302 | 0.2840 | 0.4297 | 0.4309 | 0.4925 | 0.6218 | 0.8114 |
| AOBPR | 0.3428 | 0.4005 | 0.5002 | **0.6780** | 0.4790 | 0.5375 | 0.6837 | 0.8631 |
| IRGAN | 0.3372 | 0.3957 | 0.4912 | 0.6714 | 0.4750 | 0.5327 | 0.6758 | 0.8528 |
| RNS-AS | 0.3443 | 0.3993 | 0.4992 | 0.6684 | 0.4839 | 0.5390 | 0.6832 | 0.8523 |
| AdvIR | 0.3445 | 0.3973 | 0.5018 | 0.6644 | 0.4843 | 0.5393 | 0.6839 | 0.8527 |
| SRNS | **0.3527** | **0.4093** | 0.5025 | 0.6712 | **0.4971** | **0.5505** | **0.6894** | 0.8531 |
| | 0.57% | 1.56% | -0.81% | -1.00% | 2.64% | 2.08% | 0.80% | -1.30% |

*Q2*. *Experimental results cannot be compared directly with published results due to different experimental conditions.*
**Reply.** There is no standard experimental setting that is adopted by all previous CF works. By following [28,37], we
regarded ratings with 4$\sim$5 as positive labels and evaluated with similar list lengths. We will cover more experimental
conditions in the final version.

To REVIEWER 3: *Q1*. *The concept of "hard negative samples" is used without explanation.* **Reply.** They are negative
samples with a high probability of being positive according to the model, which are hard for learning. We will elaborate
more in the final version.

*Q2*. *The analysis based on synthetic data is relatively weak, hard to justify the observation.* **Reply.** 1) Variance-based
criterion has been adopted in ML community, e.g., [8] improves stochastic optimization by emphasizing high variance
samples, and similar technique is widely used in active learning for variance reduction (see "B. Settles. Active learning
literature survey. 2010"). Here we introduce this into CF so as to filter out false negative samples. 2) The analysis on
synthetic data is motivated by the needs of a reliable measure of sample quality. 3) Experiment results on both synthetic
and real-world datasets demonstrate the effectiveness of our SRNS method.

*Q3*. *Experiment results on longer evaluation lists.* **Reply.** Please see *Q1* of REVIEWER 2.

To REVIEWER 4: *Q1*. *why SRNS is much faster than existing sampling methods.* **Reply.** SRNS can converge to better
performance (N@1) with less time (Fig. 4(a-c)). Moreover, it can be trained from scratch. For time complexity of std
computation, please see *Q3* of REVIEWER 1.

*Q2*. *For experiment on changing scoring function $r$, better to compare SRNS with RNS-AS and AdvIR.* **Reply.** Original
papers of RNS-AS and AdvIR does not consider using different $r$, thus, to be fair, we only compare SRNS with uniform
sampling to demonstrate its generality on different choices of $r$.

*Q3*. *Performance gain seems to be marginal as the number of recommended items increases.* **Reply.** 1) As in
Appendix B.4, results of both baselines and SRNS in Table 3 are tuned according to N@1 on validation set. 2) Generally
learning difficulty increases for all methods as $K$ increases.

*Q4*. *Needs to consider one candidate-based sampling method and another reinforced-based sampling method as*
*baselines.* **Reply.** [Ding et al. WWW'18] is irrelevant, as it focuses on augmenting negative samples with additional
view data, which is not available here. [Ding et al. IJCAI'19] is already compared in experiments, which is RNS-AS.

[Meta-Review · NeurIPS 2020]

The initial reviews were mixed for this paper. However, during the discussion, a certain consensus emerged regarding the value of this contribution. In particular, the reviews agree that the proposed method is simple and effective. In the proposed study, the method seems to outperform (by a small margin) others consistently. Congratulations! The main negative is the high computational and memory cost of the approach. Reviewer #1's overall score seems to have been greatly influenced by this and his assessment seems correct. While we cannot expect all new methods to be better in all aspects, the reviewers found that a proper discussion of these aspects should be provided in the main paper (and not "relegated" to the appendix). I also strongly suggest that the authors consider the reviews to improve their submission. I found the post-rebuttal comments of Rev. #2 to be particularly relevant: "as the literature has two major experimental settings (using only 4+5 star ratings as positive or using all ratings a positive), it might be worthwhile to provide results for both in the final version of the paper (maybe in an appendix)."